# Effects of small-sided games vs. simulated match training on physical performance of youth female handball players

**Rasa Mikalonytė[1], Rūtenis Paulauskas[1], Eduardo Abade [2,3], Bruno Figueira [1,4]***

**1** Educational Research Institute, Education Academy, Vytautas Magnus University, Vilnius, Lithuania,
**2** Portugal Football School, Portuguese Football Federation, Oeiras, Portugal, **3** Research Center in Sports Sciences, Health Sciences and Human Development, CIDESD, University of Tras os Montes e Alto Douro, UTAD, Vila Real, Portugal, **4** Research Center in Sports Sciences, Health Sciences and Human Development, CIDESD, University of Maia, ISMAI, Maia, Portugal

* benfigueira@hotmail.com

## Abstract

The aim of this study was to compare the effects of different Small-Sided games (SSG) formats and simulated match handball training (SMHT) on handball player's physical performance and game activity profile. Twenty-four youth female handball players (age: 16.2 ± 1.5 years) participated in this study. The study was conducted during the first part of the competitive handball season and lasted for 10 weeks with 2 sessions per week in non-consecutive days: 1 week of pretesting, 8 weeks of specific training, and 1 week of post-testing. A two-group parallel randomized, pre- to post-test design was used to compare 2 different training groups: SSG training group (n = 12) and SMHT group (n = 12). The results showed larger improvements in drop jump height, jump power, absolute and relative anaerobic alactic power and 10 m sprint performances following the SSG training compared with the SMHT (p<0.05, $\eta_p^2$ = ranging from 0.219 to 0.368). Game performance characteristics showed significant effect in SSD training in average sprint distance, total number of sprints and time between sprints (p<0.05, $\eta_p^2$ = ranging from 0.08 to 0.292). The results of this study show that handball SSGs represent an adequate in-season strategy to promote a wide range of physical adaptations with improvements in running and jumping performance. This represents important information for coaches, since SSGs develop handball players' physical profiles while replicating tactical and technical features of the game. Nevertheless, simulated match training may be judiciously used to improve players' aerobic performance.

## Introduction

Game-based training methods are by nature related to specific components of a particular sport, such as movement patterns with and without the ball, physical demands and technical requirements [1–3]. Under this scope, small-sided games (SSG) are commonly used by team sports' coaches to develop technical abilities and enhance physical performance variables [4–6]. These situations are usually developed in pitches with several sizes, involving different number of players and often played under different types of rules changes when compared to

**Data Availability Statement:** The data is only available on request, to protect the subjects' confidentiality and privacy. Interested researchers may contact the board from the Vytautas Magnus

University to request access to the data (VMU, info@vdu.lt).

**Funding:** The authors received no specific funding for this work.

**Competing interests:** The authors have declared that no competing interests exist.

the official formal match [7]. Previous studies have shown that the physical responses and technical skills requirements during SSG can be modified by manipulating several constraints, such as the size of the pitch, number of players, rules of the game and coach encouragement [1, 8, 9]. Generally, SSG with higher number of players are used to train players' tactical behaviors, while larger playing area in SSG formats are more appropriate to increase distance covered by players and high-intensity efforts [10]. On the other side, the simulated match handball training (SMHT) is fundamental, ensuring the players to experience the competitive scenario and the corresponding physical, technical and tactical requirements of match play [11]. Taking all together, specific game-based handball training is an effective training mode to improve youth players' physical profiles in several performance variables, such as vertical jump, linear sprint, repeated sprint ability and intermittent endurance [1].

When compared to match play, handball game-based training conditions may promote greater physical load and address specific players' playing position requirements [12]. Analyses of handball game demands suggest that speed, explosive strength and high-intensity intermittent running are the most important physical qualities for achieving success in high-level leagues [13]. For that purpose, SSG training is able to elicit greater improvement of explosive-like abilities when compared to high intensity interval training alone [4]. Moreover, SSG training may increase players' compliance and motivation when compared to high intensity intermittent training sessions, particularly because of high training time spent with the ball [1].

Even though game-based training strategies are recognized as useful tools for tactical, technical and physical training, leading to greater improvement in handball-specific activities such as agility, jumps, dynamic strength, and short sprints [7, 14, 15]. Time-motion analysis in women's team during match play shown that the individual mean run distance (2882 ± 1506 m) varied broadly between single field handball players, comprising 961 ± 539 m of walking, 761 ± 420 m slow running, 752 ± 484 m fast running, and 272 ± 224 m sprinting [16]. The recent development of the micro-technology has allowed the use of portable local positioning systems (LPS) to track players indoors, providing better levels of validity and reliability than the standard GPS systems [17].

Using SSG's allows coaches to increase variability between sessions, ensuring that game movement patterns are replicated, as well as the physiological and technical demands of competition under fatigue [18]. Thus, using LPS to quantify time-motion characteristics may help coaches not only to design adequate exercise strategies for different performance outcomes, but also to study the activity profiles of players [19].

Wearable tracking technologies facilitates optimization of performance by managing player workload and monitoring athlete development [20]. The results of time-motion analyses may increase the specificity of physical conditioning as they provide insight into the energy system utilization, and in some cases, specific movement patterns used throughout the course of a game [21]. However, by studying the players' performance in a simulated game, it is possible to observe and evaluate changes in their performance parameters while controlling variables such as match-ups situations and game duration. Lack of research and training monitoring methodologies in young female handball players globally limits our ability to develop adequate game-based training and research technologies.

Currently, there is only a small amount of data on the effectiveness of game-based training in handball. Nevertheless, there have been some attempts to study the effects of SSG on physical [1, 4, 8, 12], physiological [22], and tactical performance [10], showing that the manipulation of task constraints may lead to the emergence of different interpersonal relationships between players and opponents and, consequently, different physical and tactical performances. We therefore hypothesized that following SSG (2vs2, 3vs3, 4vs4) training protocol the

fitness identification measurements would indicate a greater improvement of physical performance and game activity profiles then simulated match handball training (SMHT).

## Materials and methods

### Participants

The participants were twenty-four highly trained [23] youth female handball players (age: 16.2 ± 1.5 years, stature: 168.8 ± 7.1 cm, body mass: 63.7 ± 9.5 kg; playing experience: 5.8 ± 2.3 y). The participants were from two different teams who trained under the same youth National development program, competing at youth level league U-18. The goalkeepers' positioning it is very restricted to a specific area and their positioning dynamics are different from the outfield players. Thus, 4 goalkeepers participated in the protocol, but were excluded from the analysis. The players participated in four training sessions per week, with an average duration of 90 minutes, and one official match during the weekend. The training sessions had the following structure of warm-up; handball drills, focusing on the acquisition and improvement of technical and tactical skills; small-sided handball games; and formal game. All players were healthy and were not taking any medication. The participants and their parents were informed about the research procedures, requirements, benefits and risks and their written consent was obtained before the study began. Additionally, players were informed that they were free to withdraw at any time without any penalty. Ethical approval conformed to the recommendations of the Declaration of Helsinki and was provided by the Regional Research Ethics Committee #BE-2-97.

### Design

A two-group parallel randomized, pre- to post-test design was used. 2 protocols (SSG and SMHT) were applied during 10 weeks and all players were submitted to the randomized protocol 2 sessions per week. All the players of team A were included in SSG group and all the players of team B were included in SMHT group.

### Procedures

The study was conducted during the first part of the competitive handball season (October–November) and lasted for 10 weeks, with 2 sessions per week in non-consecutive days with a duration of 33 minutes each (for a total of 20 sessions): 1 week of pretesting, 8 weeks of specific training, and 1 week of post testing [24].

A two-group parallel randomized, pre- to post-test design was used. Thus, the participants of one team were named as a SSG (n = 12) training group, and the second team was named as SMHT (n = 12) training group. To isolate the effect of the 2 training protocols, the additional fitness training sessions (e.g., technical, tactical and strength) during the 8 weeks of training were identical for both groups. To determine pretraining and post-training game activities variables within each group (SSGs and SMHT), players were divided in two sub-teams of six players. Goalkeepers took part in game training but were excluded from all analyses.

All the sessions were performed in the beginning of the training session, following a standardized 15 min warm-up based on running, dynamic stretching and ball possession drills. Similarly, sessions of both groups were performed at the same time of the day (5:00–7:00 PM) and in a similar ambient temperature (19–22º C). Coaches and players were asked to avoid intense exercise on the day before the tests and to consume their usual meal at least 3 hours before the scheduled testing time. Pre—and post—measurement were set up on the same days

of the week as for training, with first day of physical testing and another assessment of game activity.

## Protocols

**SSG.** The players of SSG training group played under 3 different formats, 2 vs. 2, 3 vs.3 and 4 vs. 4. The 3 formats were played in a random order over the 8 weeks of the study. Each SSG was played once of 10 min each, interspersed by 1 min of passive rest between the conditions, accounting for a total time of 33 min. The size of the pitch and the number of players were manipulated in an attempt to alter the intensity of the SSG and were similar to those used in previous studies [12]. The players of the 3 different formats were chosen by the head coach, considering the player's level, excluding goalkeepers and the players had to score in mini-goals (1.5 x 1 m) on each team. The 2vs.2 game was delimited by an area of 20 x 10 m (quarter regular handball court) and 3vs.3 and 4vs.4 games were played on 20 x 20 m (half regular handball court). The goal area was settled with 5m radius, and was maintained in all SSG's formats. To reduce the stoppage time and keep high intensity, no 7m penalties were awarded and in the case of the ball going off, several balls were placed around the court to ensure its replacement was provided as fast as possible and the maximal time to complete an attack before losing ball possession was present at 30 seconds. Players were asked to maintain a high pace throughout each game and to indicate their rating of perceived exertion (RPE) using the category rating 10 (CR- 10) scale modified by [25] using a standardized questionnaire.

**SMHT.** The players of SMHT protocol performed 3 bouts of 10-m of Formal Handball game interspersed with 1 minute of passive recovery, making a total of 33 minutes. The teams were chosen by the head coach, considering the player's level and the playing position, with the aim to balance the teams in terms of skill level and positions (two fullbacks, two wingers, a pivot and a center). SMHT was performed on regular handball pitch size, with regular handball rules and score in standard goals with a goalkeeper. The goalkeeper area was kept a constant in both conditions. As in the condition described above, players in both groups were asked to maintain a high pace throughout each game and to indicate their RPE using the category rating 10 (CR- 10) [25].

Table 1 provides a summary of the variables used in the study.

## Fitness identification

Jump Performance Test: Lower limb explosive power (LLEP) was assessed using a vertical drop jump (DJ) drop height = 0.20 m. Each handball player performed a DJ test with an Optojump TM device (Microgate, Bolzano, Italy) used to measure the jump height (cm) and contact time (ms) as a proxy for a muscular stretch-shortening performance. During DJ, players were advised to jump as high as possible with minimum contact time, with the hands fixated at the hips. The players repeated the test 3 times with the necessary rest and preparation. The intraclass correlation of Optojump range from 0.98 to 0.99 and standard error of measurement is 0.8.cm [26].

Computation of LLEP power in wats used the equation [27]:

$$\text{LLEP (W)} = (\text{body mass (kg)} \times 9.81 \text{ m} \cdot \text{sec}^{-2} \times \text{flight time (sec}^{-1}) \times (\text{flight time (sec}^{-1})$$
$$+ \text{contact time(sec}^{-1}))/4 \times \text{contact time (sec}^{-1}).$$

Sprint performance: Sprint ability was evaluated by a 10- and 20-m standing-start all-out run with a 2-minute rest period between all runs. The time was recorded using photocell gates (Timing-Radio Controlled; TTSport, San Marino, CA, USA) placed 0.4 m above the ground,

**Table 1. Variables of SSG and SMHT training methods.**

|  |  | SSG |  | SMHT |
| --- | --- | --- | --- | --- |
| *Variables* | *2 vs 2* | *3 vs 3* | *4 vs 4* | *6 vs 6* |
| Duration | 10 min | 10 min | 10 min | 10+10+10 min |
| Pitch size | 20 x 10 m | 20 x 20 m | 20 x 20 m | 40 x 20 m |
| Playing area (m$^2$) | 162.7* | 325.5* | 325.5* | 651.0* |
| Area per player (m$^2$) | 40.7 | 54.3 | 40.7 | 54.3 |
| $\bar{x}$ HRavg (1·min$^{-1}$) | 164.1±19.0 | 163.0±14.0 | 161.0±13.0 | 158.3±16.0 |
| $\bar{x}$ RPE (1–10 scale) | 8.6±1.4 | 8.2±1.5 | 8.3±1.2 | 6.7±1.8 |
| Goalkeepers | No | No | No | Yes |
| Rules | Applied | Applied | Applied | Regular |
| Scoring | Yes | Yes | Yes | Yes |
| Coach encouragement | Yes | Yes | Yes | Yes |

Notes:

* The goalkeeper area was not included.

Abbreviations: $\bar{x}$ = Mean values±standard deviation; m = meters; HRavg, average heart rate; RPE, rate of perceived exertion.

with an accuracy of 0.001 second. The athletes performed 3 trials for each distance and the fastest times were recorded for further analysis. The runs were performed individually by each participant. The intraclass correlation coefficient for test-retest reliability and typical error of measurement for the 10- and 20-m tests were 0.95 and 0.97, and 1.3 and 1.2%, respectively [4].

The Yo-Yo intermittent recovery test Level 1: The YYIRTL1 was used to assess players' aerobic capacity and was performed as described by Krustrup, Mohr (28). A standardized warm up prior to testing was comprised of 10 min of low-intensity running (involving basic run-throughs at an increasing tempo, dynamic stretching and change of direction activities). During testing, 20 m shuttle runs were performed at increasing velocities until exhaustion, with 10 s rest intervals of active recovery (2 x 5 m of jogging) between runs. The test was concluded when the participant failed twice to reach the front line in time (objective evaluation) or felt unable to complete another shuttle at the required speed (subjective evaluation). The distance covered was considered as the test "score". The intraclass correlation coefficient for test-retest reliability and typical error of measurement were 0.95 and 0.98, and 4.9%, respectively [28].

Handball Agility Specific Test: Agility was measured by the Handball Agility Specific Test (HAST), which was chosen because it exhibited five changes of direction at short distances, and included back and forth races, as well as lateral displacements [29]. For its realization, subject starts from cone 1 and runs in a straight line from 5m to cone 2, where it carries out lateral displacement of 3.5m to cone 3, again moves laterally 3.5m to the cone 4, runs from the back 5m to the cone 5, carries out the lateral displacement of 3.5m to the cone 3, and finally moves laterally by 3.5m to the cone 1. Two attempts have been made, with five minutes of interval between them, and the faster attempt was recorded as valid. The timing was registered from photocells arranged in the first cone. The intra-class correlation coefficient of the HAST described is 0.92 and the typical measurement error of 2.3% [4].

Margaria-Kalamen Anaerobic Alactic Power Test: The Margaria-Kalamen stair climb measure the athlete's lower body peak power [30]. The participants began the test at a starting line placed 5 meters from the first step. One timer was positioned on the 3rd step and a second timer was positioned on the 9th step. On the researcher's signal, the participant ran from the 5-meter starting mark as fast as he could up the stairway, taking the three steps at a time (3rd, 6th, 9th). The timers started recording when the participant hit the 3rd step and stopped

recording when the participant stepped on the 9th step. The average time was taken from the two timers for each trial. The participant completed 3 trials with a 20-s rest period prior to the start of each trial and the best performance time was used. The anaerobic power was measured in watts and was the product of force (weight of participant) multiplied by distance 16 (height of stairs) and acceleration due to gravity (9.81 m·sec$^{-2}$), then divided by time (sec$^{-1}$). This computation of anaerobic-alactic power in watts used the equation [31]:

$$\text{AAP (W)} = (\text{body mass(kg) x distance } (0.96 \text{ m}) \text{ x } 9.81 \text{ m} \cdot \text{sec}^{-2})/\text{time (sec}^{-1}).$$

Game activity identification: Measures of player's activity profiles were registered by triaxial accelerometers (Catapult Sports athlete tracking technology). Player load relative to playing time was used as a measure of game activity. It was measured using a portable LPS (ClearSky T6 and OptimEye S5; Catapult Innovations, Melbourne, Australia). Microsensors were placed in neoprene vests for secure attachment between the scapulae of each player, and worn underneath regular sporting attire. The triaxial accelerometer recorded players' dynamic movement in all three planes (transverse, coronal, and sagittal) at 100 Hz to calculate instantaneous PlayerLoad, which permitted a more systematic monitoring of the physical demands during the game [32]. External measures included the total load (TL) and the relative (Load·min$^{-1}$) (arbitrary units (AU)). The instantaneous PlayerLoad is a modified vector magnitude determined as the square root of the sum of the squared instantaneous rate of change in acceleration across the transverse, coronal, and sagittal planes (x, y, and z, respectively). Total distance (TD) in the game was identified in four velocity categories: stationary/walking (0–1.3 m·s$^{-1}$), jogging (1.31–3.0 m·s$^{-1}$), running (3.01–5.20 m·s$^{-1}$) and higher speed running (>5.2 m·s$^{-1}$). These speed and movement zones are similar to those used in other handball studies [16]. Data monitored by triaxial accelerometers were accumulated and processed by using software OpenField 1.18 (Catapult Innovations, Melbourne, Australia) and downloaded for further statistical calculations.

### Statistical analysis

The alpha level for all statistical tests was set *a priori* at $\alpha = 0.05$ and calculations were carried out using SPSS software V24.0 (IBM SPSS Statistics for Windows, Armonk, NY: IBM Corp.). Descriptive statistics were used to compute means and standard deviations (mean±SD), the normal distribution of the variables was assessed of samples under each condition using the Shapiro–Wilk test. Analysis of covariance (ANCOVA) was used for the evaluation of game-based training regimen differences on the parameters tested.

The statistical significance of the differences was recorded when $p < 0.05$, applying 95% confidence interval (CI). The effect size (ES) for ANCOVA was determined using partial eta squared ($\eta p^2$) and was classified as: no effect = 0 to .039, minimum = .04 to .24, moderate = .25 to .63, and strong = $\geq$ .64 [33].

### Results

Table 2 shows the descriptive statistics (mean ± SD) for physical fitness characteristics associated with the SSG and SMHT methods.

There was statistically significant improvement in SSG method for DJ (p = 0.001, minimum effect), LLEP (p = 0.024, minimum effect), absolute and relative anaerobic alactic power (p = 0.003 and p = 0.000, minimum to moderate effect) and 10 m sprint performance (p = 0.000, moderate effect). Different training formats had a strong effect on YYIRTL1 (0.001, moderate effect). In the case of applying SSG the distance decreased, while applying SMHT

**Table 2. Effect of SSG and SMHT training methods on fitness characteristics of the participants.**

| Variables | $\bar{x}$ SSG | | $\bar{x}$ SMHT | | p | Difference (95% CI) | $\eta_p^2$ |
|---|---|---|---|---|---|---|---|
| | Pre | Post | Pre | Post | | | |
| DJ (cm) | 35.3±5.7 | 37.0±4.7 | 42.4±6.1 | 41.3±6.7 | 0.001 | 5.8 (1.0–10.5) | 0.219 |
| LLEP (W) | 1008.0±167.5 | 1062.0±162.4 | 1207.0±301.8 | 1189.0±297.9 | 0.024 | 163.5 (40–367.2) | 0.232 |
| AAP (W) | 596.4±168.7 | 812.3±222.8 | 861.3±155.6 | 865.3±156.0 | 0.003 | 158.9 (32.2–285.6) | 0.248 |
| AAP (W·kg$^{-1}$) | 9.8±2.0 | 12.6±3.3 | 13.8±1.3 | 13.9±1.35 | 0.000 | 2.6 (1.0–4.3) | 0.358 |
| HAST (s) | 7.8±0.7 | 7.5±0.5 | 7.8±0.6 | 7.6±0.6 | 0.735 | 0.06 (-0.4–0.5) | 0.005 |
| 10 m (s) | 2.0±0.1 | 2.0±0.1 | 2.1±0.1 | 2.1±0.1 | 0.000 | 0.1 (0.04–0.2) | 0.050 |
| 20 m (s) | 3.6±0.2 | 3.5±0.2 | 3.6±0.2 | 3.6±0.2 | 0.358 | 0.1 (-0.11–0.2) | 0.089 |
| YYIRTL1 (m) | 1256.0±401.0 | 1122.0±354.0 | 1100.0±510.0 | 1272.0±532.0 | 0.001 | -3.38 (-386.3–379.6) | 0.368 |

Abbreviations: $\bar{x}$ = Mean values±standard deviation; p = between group-subject effect; $\eta_p^2$ = effect size; DJ = drop jump; cm = centimeters; LLEP = lower limb explosive power; W = Watts; AAP = anaerobic alactic power; W.kg$^{-1}$ = Watts per Kilogram; HAST = handball agility specific test; s = seconds; YYIRTL1 = Yo-Yo intermittent recovery test level 1; m = meters.

(p = 0.001, strong effect) the distance increased. SSG and SMHT methods had no significant effect on HAST and 20 m (p>0.05, no effect).

Effects of different game-based training on game motion characteristics of the study subjects are summarized in Table 3. No statistically significant differences were found in any component of the load (p>0.05, no effect).

Table 4 shows various game performance characteristics associated with higher speed running zone >5.21 m·s$^{-1}$.

There were significant differences between the training formats (SSG and SMHT) for mean and Average sprint distance (m) (p = 0.048, minimum effect), Total number of sprints (p = 0.021, minimum effect) and Time between sprints (s) (p = 0.000, moderate effect). In opposition, the comparison between both groups presented no significant effect on Average sprint duration (s) (p = 0.424, no effect).

## Discussion

The study aimed to compare the effects of SSG (2vs2, 3vs3 and 4vs4) and SMHT training methods on young elite female handball players' physical profile. The game motion characteristics were not influenced by either SSG or SMHT, except for those associated with sprinting at speeds above 5.21 m·s$^{-1}$). The results of this study show that handball SSGs and SMHT are

**Table 3. Effect of SSG and SMHT training methods on game motion characteristics of the participants.**

| Variables | $\bar{x}$ SSG | | $\bar{x}$ SMHT | | p | Difference (95% CI) | $\eta_p^2$ |
|---|---|---|---|---|---|---|---|
| | Pre | Post | Pre | Post | | | |
| Tl (AU) | 252.7±50.8 | 281.9±42.2 | 218.1±62.4 | 232.7±62.6 | 0.121 | -41.9 (-86.9–3.1) | 0.036 |
| L (AU) | 8.4±1.7 | 9.4±1.4 | 7.3±2.1 | 7.8±2.1 | 0.112 | -1.1 (-3.02–0.8) | 0.056 |
| TD (m) | 2460.8±202.1 | 2454.9±223.4 | 2466.5±195.3 | 2572.1±203.7 | 0.443 | 42.9 (-65.9–151.8) | 0.016 |
| TD at 0–1.30 m·s$^{-1}$ (m) | 648.8±64.6 | 664.1±95.9 | 680.0±92.8 | 693.5±105.1 | 0.917 | 3.8 (-78.4–86.0) | 0.000 |
| TD at 1.31–3.00 m·s$^{-1}$ (m) | 1159.0±179.4 | 1033.0±136.7 | 1109.0±156.7 | 1148.0±153.7 | 0.479 | 32.4 (-83.9–148.8) | 0.011 |
| TD at 3.01–5.20 m·s$^{-1}$ (m) | 605.1±164.8 | 696.7±190.2 | 633.9±146.7 | 700.8±147.9 | 0.729 | 16.5 (-115.4–148.4) | 0.003 |

Abbreviations: $\bar{x}$ = Mean values±standard deviation; p = between group-subject effect; $\eta_p^2$ = effect size; Tl = Total Load; AU = arbitrary units; L = Load min-1; TD = total distance; m = meters.

**Table 4. Effect of SSG and SMHT training methods on game motion characteristics of higher speed running (>5.21 m·s⁻¹) of the participants.**

| Variables | $\bar{x}$ SSG | | $\bar{x}$ SMHT | | p | Difference (95% CI) | $\eta_p^2$ |
|---|---|---|---|---|---|---|---|
| | Pre | Post | Pre | Post | | | |
| Asd (s) | 1.55±0.66 | 1.67±0.66 | 1.50±0.58 | 1.43±0.58 | 0.424 | -0,14 (-,067–0,38) | 0.015 |
| Asd (m) | 5.48±1.94 | 6.43±1.94 | 4.93±1.77 | 4.88±1.85 | 0.048 | -1,1 (-2,64–0,53) | 0.080 |
| Tns | 9.17±3.97 | 10.08±2.81 | 8.83±3.16 | 6.08±2.43 | 0.021 | -2,17 (-4,64–0,30) | 0.115 |
| Tbs (s) | 191.90±59.20 | 160.50±59.70 | 186.40±52.90 | 303.80±52.70 | 0.000 | 69,13 (26,79–111,46) | 0.292 |

Abbreviations: $\bar{x}$ = Mean values±standard deviation; p = between group-subject effect; $\eta_p^2$ = effect size; Asd = Average sprint duration; s = seconds; m = meters; Tns = Total number of sprints; Tbs = Time between sprints.

valid in-season strategies to promote physical profiles development of youth players. Particularly, SSG are able to promote a wide range of important physical variables in handball, such as running and jumping capacities. In parallel, SMHT may be used as an important tool to improve players' aerobic performance.

## Jumping and running performances

Research shows that the manipulation of SSG variables, such as the size of the court and the number of players, can have an adaptive effect on the player's physical fitness [1, 9]. For example, the number of sprints and running speed may vary in different small sided training formats [12] and promote different physical adaptations [13]. In other sports such as futsal, reducing the number of players may increase the frequency of technical actions such as the number of ball contacts and dribbles [34]. In the present investigation, 2x2, 3x3 and 4x4 formats were used over time, which may represent an efficient strategy to promote technical proficiency resultant from a higher number of ball contacts and increased defensive pressure from the opponents performing vertical jump shots, stride jump shots and various ball feints. Increasing the frequency of these short-term high intensity actions may be an important and time efficient strategy to reproduce match demands, combining technical, tactical and physiological workloads of the game [5]. Thus, using a concurrent approach, combining game-based drills with complementary strength training may optimize players' power development [35] required for game activities [36]. Previous studies also highlighted that the power output of the leg extensor muscles, absolute jumping power and sprint running are important neuromuscular performance characteristics for successful participation in elite levels of handball [10].

Previous research has identified differences in intensity by manipulating pitch size and keeping the number of player constant, claiming that the larger area per player elicited higher values in covered TD, RPE and heart rate [22]. It was also found that changing the number of players while holding the area per player constant do not seem to significantly affect the internal physical load parameters, however activity profile during the games differed [37]. In our study, 20x10m and 20x20m SSG formats were for 2vs2, and 4vs4, which promoted decreased area per player when compared to simulated match. Additionally, 3x3 format were played under the same field area as SMHT. Thus, physical fitness changes may occur through the manipulation of the number of players and relative area of the pitch in SSG, increasing the number of active actions and improving explosive muscular in key handball capacities such as running and jumping.

Conversely, physical fitness analysis revealed that SSG increased jumping ability and sprint speed. In fact, previous works shown that SSG's without goalkeepers increase the intensity, in

opposition in formats where goalkeepers are used where external load demands decrease [18]. Thus, the present results seems to be in line with previous reports, which have shown that the exclusion of goalkeepers in SSG's led to an increase of high-intensity actions such as percentage of time spent at high running speeds, and number of high-intensity sprints and eventually number of jumps, suggesting an attempt to adjust their positioning according to the distance of attackers to the target, aiming to decrease shooting opportunities [38].

## High intensity intermittent performance

The analysis of YYIRTL1 results showed that SMHT training increased the covered distance to a greater extent when compared to SSG intervention. This suggests that the predominant physical activity during the simulated handball game training is an intermittent high-intensity aerobic activity which may have had a specific adaptive effect on the players YYIRTL1 performance. In fact, it was already reported a strong relationship between YYIRTL1 and players' total game distance covered in young handball players, which seems to be linked to an increased ability to perform intermittent high intensity exercise for prolonged periods [39]. Concomitantly, the increase in absolute distance in the Yo-yo test in SMHT can be explained by the game principles that regulate offensive and defensive organization during formal game. While in SSG's players need to move to create opportunities to pass the ball, during SMHT players develop their action respecting the roles of specific positions, often developing activities without ball possession [7].

The profiles of game activity during handball games are determined by the complex interactions between players with and without the ball. A number of studies have been conducted to assess player workload during the games [5, 8, 13, 32]. SMHT in the regular handball pitch tend to decrease the number of active technical actions and some players may have limited roles in the game [8]. Since the players are likely to have less involvement with the ball in this format, they are required to spend time by taking less active action in offensive and defensive actions. However, the study showed that handball players are required to work active "off the ball" in high-intensity aerobic activities. Previous studies confirm that in the amount of absolute pitch space available, physical work may require players to complete an increased number of sustained runs at a variable speed [5].

## Practical applications

Coaches should be aware that game-based interventions may induce a wide range of physical adaptations. Thus, SSG should be privileged to improve short-term high intensity actions such as running and jumping performances. Additionally, simulated match training appears to be an efficient tool to improve players' overall aerobic performance. As all these outcomes are relevant to handball players physical performance, both strategies may be used according to the specific demands of in-season fixture, technical or tactical aims of the team and players' individual response throughout the annual training cycle.

## Conclusion

The results of this study show that both game-based strategies are efficient for improving handball players' physical performance during in-season. SSG were shown to have an important effect on running and jumping performances, which represents a viable strategy to combine physical, technical and tactical variables within the training units. On the other hand, SMHT may improve players' aerobic performance, which may also be considered as an interesting tool for non-starting players or players with lower match time within the team.

## Limitations

Limitation of such studies are eligible matches and psychological factors that raise the possibility that an individual athlete's individual playing style may influence player load.

On the other hand, our applied research design, and level of the players may limit the opportunity to make broader generalizations from our results to other populations. We assume that present study design could have been more powerful with a greater sample.

## Supporting information

**S1 File. Inclusivity in global research.**
(DOCX)

## Author Contributions

**Conceptualization:** Rasa Mikalonytė, Rūtenis Paulauskas, Bruno Figueira.

**Data curation:** Rūtenis Paulauskas, Bruno Figueira.

**Formal analysis:** Rūtenis Paulauskas, Bruno Figueira.

**Funding acquisition:** Bruno Figueira.

**Investigation:** Rasa Mikalonytė, Rūtenis Paulauskas, Bruno Figueira.

**Methodology:** Rasa Mikalonytė, Rūtenis Paulauskas, Eduardo Abade, Bruno Figueira.

**Project administration:** Rūtenis Paulauskas, Bruno Figueira.

**Resources:** Rūtenis Paulauskas, Bruno Figueira.

**Software:** Rūtenis Paulauskas, Bruno Figueira.

**Supervision:** Rūtenis Paulauskas, Bruno Figueira.

**Validation:** Rūtenis Paulauskas, Bruno Figueira.

**Visualization:** Rūtenis Paulauskas, Bruno Figueira.

**Writing – original draft:** Rasa Mikalonytė, Rūtenis Paulauskas, Eduardo Abade, Bruno Figueira.

**Writing – review & editing:** Rasa Mikalonytė, Rūtenis Paulauskas, Eduardo Abade, Bruno Figueira.

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
