## [Decision Letter · Decision Letter 0]

15 Mar 2022

PONE-D-22-01222Effects of small-sided games vs. simulated match training on physical performance of youth female handball playersPLOS ONE

Dear Dr. Bruno Figueira,

Thank you for submitting your manuscript to PLOS ONE. After careful consideration, we feel that it has merit but does not fully meet PLOS ONE’s publication criteria as it currently stands. Therefore, we invite you to submit a revised version of the manuscript that addresses the points raised during the review process.

We look forward to receiving your revised manuscript.

Kind regards,

Nuno Miguel Correia Leite, Ph.D.

Academic Editor

PLOS ONE

Journal Requirements:

 [NO-The funders had no role in study design, data collection and analysis, decision to publish, or preparation of the manuscript.]

[NO authors have competing interests]. 

8.  We noticed you have some minor occurrence of overlapping text with the following previous publication(s), which needs to be addressed:

- https://journals.lww.com/acsm-msse/Fulltext/2006/02000/Effects_of_an_Entire_Season_on_Physical_Fitness.24.aspx

- https://journals.lww.com/nsca-jscr/Fulltext/2015/03000/Improving_Fitness_of_Elite_Handball_Players_.36.aspx

- https://ejournals.vdu.lt/index.php/Pedagogika/article/view/1883

In your revision ensure you cite all your sources (including your own works), and quote or rephrase any duplicated text outside the methods section. Further consideration is dependent on these concerns being addressed.

Reviewers' comments:

Reviewer's Responses to Questions

**Comments to the Author**

1. Is the manuscript technically sound, and do the data support the conclusions?

Reviewer #1: Partly

Reviewer #2: Yes

2. Has the statistical analysis been performed appropriately and rigorously? 

Reviewer #1: Yes

Reviewer #2: Yes

3. Have the authors made all data underlying the findings in their manuscript fully available?

Reviewer #1: Yes

Reviewer #2: Yes

4. Is the manuscript presented in an intelligible fashion and written in standard English?

Reviewer #1: Yes

Reviewer #2: Yes

5. Review Comments to the Author

Reviewer #1: First of all, I would like to thank the authors for their work in a field that is still unexplored in handball: SSGs. With it, a range of possibilities could be opened up in terms of training methodology in relation to the physical demands and performance in this sport. It is a necessary study for the advancement of knowledge in this area.

However, I attach a document with comments and suggestions that, from my perspective, are necessary to improve the quality of this study.

Reviewer #2: Effects of small-sided games vs. simulated match training on physical performance of youth female handball players

General comments to the authors

Overall, this is a nice study that has some potential practical applications integrated with female soccer players during small-sided games in handball. The authors are commended on their efforts thus far. The study is well designed and well-written, with a great introduction proposing the usefulness of the topic and a clear outline of the research question. However, I suggest only small corrections and the authors should update the recent references about small-sided games. These corrections and studies will allow improving the manuscript.

Abstract

Instead of The results showed larger improvements in drop jump (cm)

(p=0.001, ηp2=0,219), jump power (w) (p=0.024, ηp2=0,232), absolute and relative

anaerobic alactic power (W and W•kg -1 ) (p=0.003, ηp2=0,248 and p=0.000,

ηp2=0,358 respectively) and 10 m sprint performance (p=0.000) in SSG group. SMHT

group improved Yo-Yo intermittent recovery test Level 1 distance (p=0.0001,

ηp2=0,368) to a greater extent

you should use this sentence clear and shortly

The results showed larger improvements in drop jump height, jump power, absolute and relative anaerobic alactic power and 10 m sprint performances following the SSG training compared with the SMHT (p ≤ 0.05, ηp2=ranging from 0.219 to 0.368).

Instead of drop jump (cm), you should use drop jump height.

Similarly performance responses, Game performance characteristics should be written.

Introduction section

Page 3, Line 48: The authors should add recent references about small-sided games

Arslan, E., Kilit, B., Clemente, F. M., Soylu, Y., Sögüt, M., Badicu, G., ... & Murawska-Ciałowicz, E. (2021). The Effects of Exercise Order on the Psychophysiological Responses, Physical and Technical Performances of Young Soccer Players: Combined Small-Sided Games and High-Intensity Interval Training. Biology, 10(11), 1180.

Page 3, Line 50: The authors should add recent references about small-sided games

Arslan, E.; Kilit, B.; Clemente, F.M.; Murawska-Ciałowicz, E.; Soylu, Y.; Sogut, M.; Akca, F.; Gokkaya, M.; Silva, A.F. Effects of Small-Sided Games Training versus High-Intensity Interval Training Approaches in Young Basketball Players. Int. J. Environ. Res. Public Health 2022, 19, 2931. https://doi.org/10.3390/ijerph19052931

Page 3 and 4: The authors should add recent references about small-sided games in handball and also discussion section

Jurišić, M. V., Jakšić, D., Trajković, N., Rakonjac, D., Peulić, J., & Obradović, J. (2021). Effects of small-sided games and high-intensity interval training on physical performance in young female handball players. Biology of Sport, 38(3), 359.

Methodology

Page 5, Line 117: it should be (e.g., technical, tactical and strength)

Page 6, Line 119: it should be within each group (SSGs and SMHT), players were divided into

Page 6, Line 125: it should be (19–22 °C)

Protocols

Page 6, Line 135: which one is the your style you have to make a decision

10-min or 33 min. Please be careful throughtout the article

Page 6, Line 137: Instead of small-sided games, it should be SSG

Page 7, Line 146: Instead of rate of perceived exertion, it should be rating of perceived exertion

SMHT

Page 7, Line 157: Instead of to indicate their rate of perceived exertion (RPE), it should be to indicate their RPE

Fitness identification

Page 7, Line 164: you do not need (W)

Page 9, Line 200: Margaria-Kalamen AAP Test ??? what are they AAP???

Results section

This section is well designed and well-written

Discussion section

This section is well designed and well-written. However, the authors should add limitations and strengths of their article.

Tables

This section is well designed and well-shown

6. PLOS authors have the option to publish the peer review history of their article (what does this mean?). If published, this will include your full peer review and any attached files.

Reviewer #1: No

Reviewer #2: No

---

## [Author Response · Author response to Decision Letter 0]

2 Apr 2022

Dear editor and reviewers

Thank you very much for the opportunity to re-submit the manuscript, as well as for all the valuable and helpful comments and suggestions. We do believe that the paper has significantly improved after this revision. We have modified the manuscript according to all comments and suggestions raised by the reviewers. The answers are presented in RED through manuscript and in GREEN in the review file.

Reviewer 1

Specific and general comments (by section) ABSTRACT

• Set the eta square symbol appropriately

• Thank you for the feedback. Changed accordingly.

• Remove units of measurement from results

• Thank you for the feedback. Changed accordingly.

• Delete the subjective consideration ‘a greater extent’ from results

• Thank you for the feedback. Changed accordingly.

INTRODUCTION

• The introduction presents interesting content about small-sided games, such as the objectives pursued with their implementation, the results associated with handball or the advantages of their use. However, in the first paragraph, it would be useful to briefly introduce the concept itself. That is, what are the SSGs and what are their main characteristics. This first paragraph, as a whole, seems to me to be quite adequate. However, I would split it in two (line 55) to talk, in its second part (new paragraph 2), about the comparison between SSG and SMHT. Previously, in paragraph 1, the concept of SMHT should be introduced and developed as it has been done with SSGs.

• Thank you for the feedback. The section was reworded.

• In relation to the previous point, the published scientific evidence on SSG and SMHT in handball should be incorporated in a third paragraph and, therefore, develop the idea already expressed in lines 80-86. Thus, it would be possible to check, for example, how the SSG would evaluate changes in physical performance in conditions without competitive anxiety or, on the contrary, whether or not the SMHT are effective on the physical performance of male and female handball players. Therefore, developing the link between SSG and physical performance in handball (HR, external load, etc.).

• Thank you for the feedback. The section was reworded. Also, the reference to competitive anxiety was removed. Although we recognize its importance in sports performance, the rationale of the study does not explore this issue. 

• With regard to the relationship between SSG - SMHT and game activity profile (2nd part of the ‘Introduction’ section), the information on the inclusion of LPS systems and their differentiation from GPS seems to me to be timely. However, I do not fully appreciate the relationship between what can be measured by this technology, the SSG/SMHT and the game activity profile. The question that any reader could ask would be: do SSGs, based on the information extracted from LPS systems, really enhance specific game activity profiles? It would therefore be necessary to link the last two aspects with the SSGs.

• Thank you for the feedback. The information was added.

MATERIAL AND METHODS

Sample:

• Please add the average handball experience of the players (important to contextualise the effects of SSGs) and whether the players had previous contact with this type of training.

• Thank you for the feedback. The information was added.

• Please justify the exclusion criteria for goalkeepers (line 95). And add the number of them.

• Thank you for the feedback. The information was added.

• Could the authors define accurately the elite level of the players: national team, youth league (U-18) or other? And if they are part of a talent detection and development programme, please could be specified it?

• Thank you for the feedback. The information was added.

Study design:

• In terms of study design and participants, were the two randomised groups performed in both teams? That is, was there a control group (SMTH) and an experimental group (SSG) in both team 1 and team 2, or, conversely, were all players in one team a control group and those in the other team an experimental group? This aspect seems to become clearer with the implementation of the pre- test and post-test (lines 118-120), but specify it for the whole intervention. And, the groups were counterbalanced besides randomized?

• Thank you for the feedback. We understand and recognize the pertinence of your question. However, due to training organization constraints (i.e. technical staff training model) the establishment of control groups was not possible. As you may understand, the intervention of researchers is somehow limited when data is collected in real world training scenarios. The information about teams’ balance was added.

• Furthermore, in order to make your research more rigorous, please add the following reference to support the duration of the training programme and the frequency of sessions per week.

- Hammami, A., Gabbett, T. J., Slimani, M., & Bouhlel, E. (2018). Does small-sided

games training improve physical-fitness and specific skills for team sports? A systematic review with meta-analysis. Journal of Sports Medicine and Physical Fitness, 58(10), 1446-1455.

• Thank you for the feedback. The reference was added.

Methodology (I would call this section ‘Procedures’ and I would include all the sub- sections (training protocols, schedule, fitness test, etc.)

• Thank you for the feedback. Changed accordingly.

• Protocols – SSG. Reference no. 4 (line 138) concerning the size of the pitch should be deleted or replaced by another reference concerning handball. In the study by Hill-Hass (2009), which deals with football, it has little or nothing to do with the handball field dimensions.

• Thank you for the feedback. Changed accordingly.

• Protocols – SSG. Please define the dimensions of the goal area in the SSGs. It is important in relation to the distance to be covered by the players.

• Thank you for the feedback. The information was added.

• Protocols – SSG. The following comment is a personal doubt as a handball player, coach and teacher: Do the authors think that the introduction of mini-goals without goalkeeper is a more effective modification in a SSG than playing with goalkeeper in an official goal (3mx2m)?

• Thank you for your question. We do understand the rationale of your question and we strongly believe that manipulating such constraints would have an important impact on the players’ behavior. However, current research in handball only allows us to speculate. Due to the particular characteristics of handball, we think that removing GK from the equation would have a greater impact on players’ performance (even on their motivation) than playing with GK.

• Protocols. Could the authors explain or justify active rest in SSGs and yet passive recovery in SMHTs?

• Thank you for the feedback. Actually, this is a mistake Rest between SSGs was passive. The information was corrected in the manuscript. 

• Protocols. SMTH. Were the players chosen by the coach in the SSGs, as was the case in the SMTHs? If this action was carried out with the aim of equalising the technical-tactical level of the team-groups, it should also have been carried out in the SSGs.

• Thank you for the feedback. The information was added.

• Fitness tests. For physical assessments, the inter-test coefficient of variation (CV) and an intra-class correlation coefficient (ICC) should be provided as informative indices of inter-test variability and consistency of observers' measurements, respectively. The authors provide it in all tests except YYIRTL1 and SJ. Please provide them in these tests as well.

• Thank you for the feedback. However, SJ was not used to assess jumping performance. Results section was updated.

• Fitness tests. For the measurement of jumping ability, why was the DJ used instead of the CMJ or unilateral CMJ?

• Thank you for your question. We do recognize the relevance of CMJ as a useful index of the muscular ability to generate force. However, the rationale for selecting DJ was related to the importance of reactive strength in handball, as players are frequently required to perform jumping actions with short ground contact times.

RESULTS

• In the footnotes to the tables, make sure that you only add the abbreviations in order of appearance (left-right and top-bottom). The footnote to Table 4 is not complete. Do not include redundant information such as 'The value expressed as mean and standard deviation (SD) in both game-based training (SSG) and (SMHT) groups', but insert the abbreviation (X ± SD) in the table itself.

• Thank you for the feedback. Changed accordingly.

• Lines 254-256 can be summarised as 'No statistically significant differences were found in any component of the load'.

• Thank you for the feedback. The sentence was reworded.

• Provide a suitable format of Table 4 

• Thank you for the feedback. Changed accordingly.

DISCUSSION

• According to the objective of the study connected to the game activity profile, it would be appropriate to add in the first paragraph - summary of the ‘Discussion’ section the following sentence: ‘the game motion characteristics were not influenced by either SSG or SMHT, except for those associated with sprinting at speeds above 5.21 m·s-1).”

• Thank you for the feedback. The sentence was added.

• Throughout the manuscript, but especially in the ‘Discussion’ section, the statement relating the number of players and factors associated with internal load, such as RPE and HR (lines 280-282), raises my doubts. Firstly, because the results provided in Table 1 are merely descriptive (X ± SD) and have not been statistically analysed. And secondly, because only considering this descriptive information, for example, the HR decreased when the number of players per team was reduced from 4 to 3, contrary to this statement. However, the biggest drawback is the one discussed in point 1 of this comment. On the other hand, the total distance covered has not been analysed in terms of the type of small-sided game but in the comparison between SSGs and SMHTs. The distance is likely to be greater the fewer players there are, but it needs to be demonstrated.

Therefore, the case of Belka et al. (2009) is not the case in this study.

• Thank you for the feedback. We do understand and agree with the reviewer perspective. As so, the sentence/ref was removed.

• On the other hand, the total distance covered in the YYRTL1 test is analysed, being greater after the SMTH protocol than after the SSG protocol (Table 2). However, this difference in physical performance does not translate to the game activity profile as there is no difference (p = 0.443) between the total distance recorded in the SMHT and the SSG. And that should be the main aspect to discuss with regard to aerobic capacity in handball. That is, why is there a difference in the tests but not in the training? The idea of lines 310-313, more elaborated by comparing the previous results, would be the right one to develop this explanation.

• Thank you for the feedback. The idea underlined on lines 310-313 was updated a new reference was added.

GENERAL COMMENT: 

In this section it is not sufficiently clear why the following results have been obtained: (1) SSG enhance jumping ability and sprint speed; (2) SMHT favour the development of aerobic capacity (CAUTION, in test not in relation to game activity profile). For example, on the SSG point, in addition to reducing the number of players and the size of the pitch, it would not be possible to investigate the application of adapted rules (e.g. inclusion or not of the goalkeeper - Table 1) or the disappearance of playing positions? It is another matter whether the corresponding justifications for all this, which indeed focus on the aspect of physical and conditional demands, are well justified (e.g. ‘These short-term high intensity actions may impose higher physiological loads and also allow stimuli for muscle power development’ lines 287-288). Furthermore, in the subheadings of the 'Discussion' section, in my opinion, I would not refer to the possible explanations, but rather to the main results: increased jumping power, increased speed and increased alactic anaerobic power. Therefore, a possible structure of this section would be, results→explanations related to handball→ justification with data extracted from the present study. Consider this study to improve this section:

- Clemente, F. M., Afonso, J., & Sarmento, H. (2021). Small-sided games: An umbrella review of systematic reviews and meta-analyses. PLoS One, 16(2), e0247067.

• Thank you for your suggestions. The discussion section was reformulated. Also, two main subheadings were considered in the new structure: jumping/running performances and high intensity intermittent performance.

PRACTICAL APPLICATIONS

• The practical applications are general conclusions of the study, but not a list of specific practical actions that coaches or physical trainers can implement in handball training. Practically, it is the same content as in paragraph 1 of the ‘Discussion’ section.

• Thank you for your feedback. This section was reworded.

CONCLUSIONS

• The conclusion is a replication of the first paragraph of the ‘Discussion’ and ‘Practical Applications’ sections. This part of the manuscript should add value to the results found.

• Thank you for your feedback. This section was reworded.

REFERENCES

• Please check the format of the scientific journals in the ‘References’ section. You may not use abbreviations for some journals (i.e.: J Strength Cond Res.) and the full name for others (i.e.: European journal of sport science.).

• Thank you for the feedback. Changed accordingly.

MINNOR COMENTS

• Line 119. Delete ‘to’

• Thank you for the feedback. Changed accordingly.

• Line 125. Add the symbol ‘oC’

• Thank you for the feedback. Changed accordingly.

• Line 141. Delete ‘court size’

• Thank you for the feedback. Changed accordingly.

• Furthermore, check especially for any missing or incorrect spelling, capitalisation,

etc.

• Thank you for the feedback. Changed accordingly.

Reviewer 2

General comments to the authors

Overall, this is a nice study that has some potential practical applications integrated with female soccer players during small-sided games in handball. The authors are commended on their efforts thus far. The study is well designed and well-written, with a great introduction proposing the usefulness of the topic and a clear outline of the research question. However, I suggest only small corrections and the authors should update the recent references about small-sided games. These corrections and studies will allow improving the manuscript.

• Thank you for the feedback. 

Abstract

Instead of The results showed larger improvements in drop jump (cm) (p=0.001, ηp2=0,219), jump power (w) (p=0.024, ηp2=0,232), absolute and relative anaerobic alactic power (W and W·kg -1 ) (p=0.003, ηp2=0,248 and p=0.000,ηp2=0,358 respectively) and 10 m sprint performance (p=0.000) in SSG group. SMHT group improved Yo-Yo intermittent recovery test Level 1 distance (p=0.0001,ηp2=0,368) to a greater extent

you should use this sentence clear and shortly

The results showed larger improvements in drop jump height, jump power, absolute and relative anaerobic alactic power and 10 m sprint performances following the SSG training compared with the SMHT (p ≤ 0.05, ηp2=ranging from 0.219 to 0.368). 

• Thank you for the feedback. Changed accordingly.

Instead of drop jump (cm), you should use drop jump height.

• Thank you for the feedback. Changed accordingly.

Similarly performance responses, Game performance characteristics should be written.

• Thank you for the feedback. Changed accordingly.

Introduction section

Page 3, Line 48: The authors should add recent references about small-sided games

Arslan, E., Kilit, B., Clemente, F. M., Soylu, Y., Sögüt, M., Badicu, G., ... & Murawska-Ciałowicz, E. (2021). The Effects of Exercise Order on the Psychophysiological Responses, Physical and Technical Performances of Young Soccer Players: Combined Small-Sided Games and High-Intensity Interval Training. Biology, 10(11), 1180.

• Thank you for the feedback. The reference was added.

Page 3, Line 50: The authors should add recent references about small-sided games 

Arslan, E.; Kilit, B.; Clemente, F.M.; Murawska-Ciałowicz, E.; Soylu, Y.; Sogut, M.; Akca, F.; Gokkaya, M.; Silva, A.F. Effects of Small-Sided Games Training versus High-Intensity Interval Training Approaches in Young Basketball Players. Int. J. Environ. Res. Public Health 2022, 19, 2931. https://doi.org/10.3390/ijerph19052931

• Thank you for the feedback. The reference was added.

Page 3 and 4: The authors should add recent references about small-sided games in handball and also discussion section 

Jurišić, M. V., Jakšić, D., Trajković, N., Rakonjac, D., Peulić, J., & Obradović, J. (2021). Effects of small-sided games and high-intensity interval training on physical performance in young female handball players. Biology of Sport, 38(3), 359.

• Thank you for the feedback. The reference was added.

Methodology

Page 5, Line 117: it should be (e.g., technical, tactical and strength)

• Thank you for the feedback. Changed accordingly.

Page 6, Line 119: it should be within each group (SSGs and SMHT), players were divided into

• Thank you for the feedback. Changed accordingly.

Page 6, Line 125: it should be (19–22 °C)

• Thank you for the feedback. Changed accordingly.

Protocols

Page 6, Line 135: which one is the your style you have to make a decision 10-min or 33 min. Please be careful throughtout the article

• Thank you for the feedback. Changed accordingly.

Page 6, Line 137: Instead of small-sided games, it should be SSG

• Thank you for the feedback. Changed accordingly.

Page 7, Line 146: Instead of rate of perceived exertion, it should be rating of perceived exertion

• Thank you for the feedback. Changed accordingly.

SMHT

Page 7, Line 157: Instead of to indicate their rate of perceived exertion (RPE), it should be to indicate their RPE

• Thank you for the feedback. Changed accordingly.

Fitness identification

Page 7, Line 164: you do not need (W)

• Thank you for the feedback. Changed accordingly.

Page 9, Line 200: Margaria-Kalamen AAP Test ??? what are they AAP???

• Thank you for the feedback. Changed accordingly.

Results section

This section is well designed and well-written

• Thank you for the feedback. 

Discussion section

This section is well designed and well-written. However, the authors should add limitations and strengths of their article.

• Thank you for the feedback. The section was added.

Tables

This section is well designed and well-shown

• Thank you for the feedback. Changed accordingly.

---

## [Decision Letter · Decision Letter 1]

27 Jun 2022

PONE-D-22-01222R1Effects of small-sided games vs. simulated match training on physical performance of youth female handball playersPLOS ONE

Dear Dr. Figueira,

Thank you for submitting your manuscript to PLOS ONE. After careful consideration, we feel that it has merit but does not fully meet PLOS ONE’s publication criteria as it currently stands. Therefore, we invite you to submit a revised version of the manuscript that addresses the points raised during the review process.

Your revised manuscript has been assessed again by the two reviewers from the previous round. The reviewers acknowledge that the manuscript has improved, but reviewer 1 has some remaining concerns which must be addressed before the manuscript can be deemed suitable for publication. Please see the attached document for the full comments.

We look forward to receiving your revised manuscript.

Kind regards,

Joseph Donlan

Editorial Office

PLOS ONE

Reviewers' comments:

Reviewer's Responses to Questions

**Comments to the Author**

1. If the authors have adequately addressed your comments raised in a previous round of review and you feel that this manuscript is now acceptable for publication, you may indicate that here to bypass the “Comments to the Author” section, enter your conflict of interest statement in the “Confidential to Editor” section, and submit your "Accept" recommendation.

Reviewer #1: All comments have been addressed

Reviewer #2: All comments have been addressed

2. Is the manuscript technically sound, and do the data support the conclusions?

Reviewer #1: Partly

Reviewer #2: Yes

3. Has the statistical analysis been performed appropriately and rigorously? 

Reviewer #1: Yes

Reviewer #2: Yes

4. Have the authors made all data underlying the findings in their manuscript fully available?

Reviewer #1: Yes

Reviewer #2: Yes

5. Is the manuscript presented in an intelligible fashion and written in standard English?

Reviewer #1: Yes

Reviewer #2: Yes

6. Review Comments to the Author

Reviewer #1: I am grateful to the authors for their efforts in making the suggested corrections and changes. However, there are still some issues that need to be clarified/clarified before accepting the manuscript. In the attached file you can find each of them. Please reference the changes noticeably in the manuscript.

Reviewer #2: Overall, this is a nice study that has some potential practical applications integrated with female soccer players during small-sided games in handball. The authors are commended on their efforts thus far. Acceted

7. PLOS authors have the option to publish the peer review history of their article (what does this mean?). If published, this will include your full peer review and any attached files.

Reviewer #1: No

Reviewer #2: No

---

## [Author Response · Author response to Decision Letter 1]

5 Jul 2022

Dear editor and reviewers

Thank you very much for the opportunity to re-submit the manuscript, as well as for all the valuable and helpful comments and suggestions. We do believe that the paper has significantly improved after this revision. We have modified the manuscript according to all comments and suggestions raised by the reviewers. The answers are presented in RED through manuscript and in GREEN in the review file.

INTRODUCTION

• In relation to the previous point, the published scientific evidence on SSG and SMHT in handball should be incorporated in a third paragraph and, therefore, develop the idea already expressed in lines 80-86. Thus, it would be possible to check, for example, how the SSG would evaluate changes in physical performance in conditions without competitive anxiety or, on the contrary, whether or not the SMHT are effective on the physical performance of male and female handball players. Therefore, developing the link between SSG and physical performance in handball (HR, external load, etc.).

Reviewer#: Although the link to anxiety status has been removed, the most relevant scientific evidence in relation to SSGs has not been incorporated. As the authors state in lines 96-97, there are studies, albeit few, that develop the link between SSGs and physical performance (or other factors) in handball. For this reason, the ‘Introduction’ section cannot be considered complete until such information is included. Please add a paragraph to this effect.

• Thank you for the feedback. The paragraph was added. Line 92-97

• With regard to the relationship between SSG - SMHT and game activity profile (2nd part of the ‘Introduction’ section), the information on the inclusion of LPS systems and their differentiation from GPS seems to me to be timely. However, I do not fully appreciate the relationship between what can be measured by this technology, the SSG/SMHT and the game activity profile. The question that any reader could ask would be: do SSGs, based on the information extracted from LPS systems, really enhance specific game activity profiles? It would therefore be necessary to link the last two aspects with the SSGs.

Reviewer#: done. Add the reference in line 84.

• Thank you for the feedback. The reference was added. Line 80.

MATERIAL AND METHODS

Sample:

• Please add the average handball experience of the players (important to contextualise the effects of SSGs) and whether the players had previous contact with this type of training.

 Reviewer#: Information on the background of the players regarding their participation in this type of training (SSGs) has not been included. It is important to know if there was a selection bias in the study sample in relation to experience.

• Thank you for the feedback. The information was added. Line 111-113.

Study design:

• In terms of study design and participants, were the two randomised groups performed in both teams? That is, was there a control group (SMTH) and an experimental group (SSG) in both team 1 and team 2, or, conversely, were all players in one team a control group and those in the other team an experimental group? This aspect seems to become clearer with the implementation of the pre- test and post-test (lines 118-120), but specify it for the whole intervention. And, the groups were counterbalanced besides randomized?

Reviewer#: In my opinion, assigning each of the players to a group, control or experimental, should be a prior task that every researcher o technical staff member can perform. And even more so, with an affordable number of participants. If the authors have not defined this aspect, add the sample randomisation method for the formation of the experimental and control groups. On the other hand, I cannot find the information on the teams’ balance.

• Thank you for the feedback. The information was added in design section. Line 124-125

Procedures

• Protocols – SSG. Please define the dimensions of the goal area in the SSGs. It is important in relation to the distance to be covered by the players.

Reviewer#: The regulation dimensions of a handball area imply a length of approximately 16 metres from the end line. If, as the authors state, the dimensions of the area were maintained for the different game formats: (i) in the 3vs3 and 4vs4 format played on a 20x20m court the playing space was excessively reduced; (ii) and furthermore, in the 2vs2 format (20x10m) it was materially impossible to introduce a goal area of these dimensions.

• Thank you for the feedback. The information was added. Line160-161.

• Fitness tests. For physical assessments, the inter-test coefficient of variation (CV) and an intra-class correlation coefficient (ICC) should be provided as informative indices of inter-test variability and consistency of observers' measurements, respectively. The authors provide it in all tests except YYIRTL1 and SJ. Please provide them in these tests as well.

Reviewer#: Both coefficients (CV and ICC), which are still not provided, should be included in the ‘Method’ section as the authors themselves do for the other fitness tests (e.g. lines 198-199 and 218-129).

• Thank you for the feedback. The information was added. Line188-190 and 210-212.

RESULTS

• Lines 254-256 can be summarised as 'No statistically significant differences were found in any component of the load'.

Reviewer#: Add ‘p > 0.05’ at the end of the sentence. Remember it is a ‘Result’ section

• Thank you for the feedback. Changed accordingly. Line 278

DISCUSSION – GENERAL COMMENT:

Reviewer#: I am unable to find the differences between the ‘Discussion’ section of the original submission and this one in the second. While the changes have been made to the subtitles, I am unable to identify the changes made (please highlight the changes in red or activate the tracked changes function in your document to detect them. Also, add in the ‘authors' response’ the lines in the new manuscript where these changes have been made. Do this for the whole document). So, I copy my concerns again: In this section it is not sufficiently clear why the following results have been obtained: (1) SSG enhance jumping ability and sprint speed; (2) SMHT favour the development of aerobic capacity (CAUTION, in test not in relation to game activity profile). For example, on the SSG point, in addition to reducing the number of players and the size of the pitch, it would not be possible to investigate the application of adapted rules (e.g. inclusion or not of the goalkeeper - Table 1) or the disappearance of playing positions? It is another matter whether the corresponding justifications for all this, which indeed focus on the aspect of physical and conditional demands, are well justified (e.g. ‘These short-term high intensity actions may impose higher physiological loads and also allow stimuli for muscle power development’ lines 287-288). Finally, I consider that the reference mentioned (Clemente et al. 2021) is more than timely to vertebrate this section.

• Thank you for the feedback. The information was added. Line 309-314, line 328-335, and line 345-349.

REFERENCES

• Modify the reference in line 350 according to journal style 

• Thank you for the feedback. Changed accordingly. Line 359

MINNOR COMENTS

• If you want to use abbreviations in the ‘Conclusion’ section, please do so in all the terms included (line 367)

• Thank you for the feedback. Changed accordingly. Line 375.

---

## [Decision Letter · Decision Letter 2]

11 Aug 2022

Effects of small-sided games vs. simulated match training on physical performance of youth female handball players

PONE-D-22-01222R2

Dear Dr. Figueira,

We’re pleased to inform you that your manuscript has been judged scientifically suitable for publication and will be formally accepted for publication once it meets all outstanding technical requirements.

Kind regards,

Ersan Arslan, Ph.D.

Guest Editor

PLOS ONE

Additional Editor Comments (optional):

Reviewers' comments:

Reviewer's Responses to Questions

**Comments to the Author**

1. If the authors have adequately addressed your comments raised in a previous round of review and you feel that this manuscript is now acceptable for publication, you may indicate that here to bypass the “Comments to the Author” section, enter your conflict of interest statement in the “Confidential to Editor” section, and submit your "Accept" recommendation.

Reviewer #1: All comments have been addressed

Reviewer #2: All comments have been addressed

2. Is the manuscript technically sound, and do the data support the conclusions?

Reviewer #1: Yes

Reviewer #2: Yes

3. Has the statistical analysis been performed appropriately and rigorously? 

Reviewer #1: Yes

Reviewer #2: Yes

4. Have the authors made all data underlying the findings in their manuscript fully available?

Reviewer #1: Yes

Reviewer #2: Yes

5. Is the manuscript presented in an intelligible fashion and written in standard English?

Reviewer #1: Yes

Reviewer #2: Yes

6. Review Comments to the Author

Reviewer #1: The authors have made a great effort in editing the manuscript, improving it considerably, especially in the argumentation and justification of the results obtained.

In my opinion, the paper represents an advance in the knowledge that SSGs and SMHTs can provide at a physical level in the training/performance process in women's handball. I encourage the authors to progress in this direction.

My sincere congratulations

Reviewer #2: The reviewer would like to thank the authors for their work and efforts in trying to improve sports science knowledge

7. PLOS authors have the option to publish the peer review history of their article (what does this mean?). If published, this will include your full peer review and any attached files.

Reviewer #1: No

Reviewer #2: No

---

## [Editor Report · Acceptance letter]

31 Aug 2022

PONE-D-22-01222R2 

Effects of small-sided games vs. simulated match training on physical performance of youth female handball players 

Dear Dr. Figueira:

I'm pleased to inform you that your manuscript has been deemed suitable for publication in PLOS ONE. Congratulations! Your manuscript is now with our production department. 

Kind regards, 

on behalf of

Dr. Ersan Arslan 

Guest Editor

PLOS ONE